# On the Limits of End-to-End Learning: Why Statistical Features Dominate in Hyper-Fragmented Battery Prognostics

## Abstract

The estimation of State-of-Health (SOH) for Electric Vehicle (EV) batteries from real-world operational data is a critical industrial challenge, primarily due to the "hyper-fragmented" nature of the data. Recent studies have shown that complex hybrid deep learning models, which rely on extensive hand-crafted features, can achieve high performance on this data. However, a fundamental question remains unanswered: Can the prevailing end-to-end learning paradigm autonomously learn effective representations from such fragmented raw signals without the aid of domain-specific feature engineering? This paper directly investigates this question through a rigorous comparative study. We contrast two starkly different paradigms: (1) a traditional machine learning approach using a CatBoost model on a novel, compact 4-dimensional statistical feature vector derived from lifetime vehicle signals, and (2) a pure end-to-end approach employing a comprehensive suite of seven advanced deep learning architectures, including 1D-CNNs, LSTMs, and Transformers. Our results reveal a significant performance disparity: the feature engineering approach achieves a robust $R^2$ of approximately 0.80, whereas the best-performing, statistically validated end-to-end model only reaches an estimated $R^2$ of 0.12. This work provides compelling empirical evidence that for high-noise, hyper-fragmented industrial time-series tasks, the information encoded in simple statistical features provides a more effective signal for prognostics than representations learned by current end-to-end architectures, highlighting a critical boundary for their application.

## 1 Introduction

The successful application of deep learning to real-world industrial systems often requires bridging the gap between models trained on curated, laboratory-like data and the noisy, stochastic nature of operational environments. This "lab-to-real" challenge is particularly pronounced in the field of battery prognostics, where accurately estimating the State-of-Health (SOH) is paramount for the safety and reliability of Electric Vehicles (EVs) (Wu et al., 2024; Massaoudi et al., 2024; Hu et al., 2025; 2024). While the prevailing paradigm in representation learning suggests that end-to-end models should autonomously learn effective features from raw data (Fu et al., 2022; Gao et al., 2024), the validity of this hypothesis under severe real-world data constraints, such as data deficiency (Wang et al., 2025), is not well-established.

This paper investigates this fundamental question using the large-scale 'IVST-EV' operational dataset, which was recently introduced and made public by (Liu et al., 2025). A key challenge of this dataset, which we term "hyper-fragmentation", is that the time-series data consists of millions of short, disconnected segments, a property that invalidates many standard modeling assumptions. In their foundational work, Liu et al. (2025) demonstrated that a sophisticated multi-modal *hybrid* deep learning framework—one that relies on an extensive pipeline of engineered features—can achieve high SOH estimation accuracy. The success of this feature-rich hybrid model, however, reveals that the final performance is a result of a *combination* of an extensive feature engineering pipeline and a deep learning architecture. This makes it difficult to disentangle the true source of the predictive power and motivates us to ask two fundamental questions: 1) To what extent is feature engineering

a *prerequisite* for success? 2) Can modern end-to-end architectures, when isolated, autonomously learn effective representations directly from such challenging raw signals?

Recent reviews on explainable artificial intelligence (AI) underscore the importance of answering such questions to build trust in safety-critical systems (Wang & Chen, 2024). Therefore, in this challenging context, we conduct a head-to-head comparison to answer our central research question: **Under the severe constraints of hyper-fragmentation, can end-to-end deep learning architectures learn effective representations for SOH estimation that surpass those from carefully engineered statistical features?** We approach this by systematically evaluating two competing methodologies: a traditional machine learning pipeline built upon a novel, compact statistical feature vector, and an extensive suite of advanced deep learning models.

Our findings present a stark, counter-intuitive result. We demonstrate that the traditional feature engineering approach not only performs better but does so by a remarkably large margin, achieving an $R^2$ score of $\approx 0.80$ while the best deep learning counterpart only reaches $\approx 0.12$. This work makes the following contributions:

- We provide a rigorous, large-scale empirical analysis **stress-testing the pure end-to-end learning paradigm** on a noisy and hyper-fragmented real-world industrial dataset.
- We propose a novel 4-dimensional statistical feature vector that is robust to data fragmentation and proves highly effective for capturing battery degradation signals (Wen et al., 2024).
- We present a conclusive finding that, for this task, a simple engineered representation decisively outperforms complex learned representations, **serving as a critical data point on the limitations and failure boundaries of the end-to-end paradigm** in certain industrial settings.

## 2 RELATED WORK

Our research is positioned at the intersection of two distinct paradigms for data-driven prognostics: end-to-end deep learning on raw sequential data, and traditional machine learning on engineered features.

### 2.1 DEEP LEARNING FOR SOH PROGNOSTICS ON STRUCTURED DATA

The estimation of battery SOH has become a prominent benchmark task for advanced deep learning models. A significant body of literature has demonstrated the power of Recurrent Neural Networks (RNNs) and their variants, such as LSTMs and GRUs, to model the temporal dependencies in battery degradation signals (Zhang et al., 2018; Li et al., 2019a; Goodfellow et al., 2016). Hybrid models, such as CNN-LSTMs, use convolutional layers to extract local features before feeding them into a recurrent network (Ren et al., 2021; Tian et al., 2022; Chemali et al., 2018). More recently, the success of Transformer architectures in sequence modeling has been translated to the battery domain, with models incorporating self-attention mechanisms showing state-of-the-art performance in capturing long-term dependencies (Song et al., 2023; Hannan et al., 2023). Even large language model frameworks are being explored for their potential in this domain (Yunusoglu et al., 2025).

However, a crucial, unifying characteristic of these successful applications is their reliance on well-structured, laboratory-generated datasets (e.g., NASA(Saha & Goebel, 2007), CALCE(Birkl, 2017), OxfordBirkl (2017)). These datasets feature clean, full charge-discharge cycles, providing a high signal-to-noise ratio and consistent temporal patterns (Zhang & Li, 2022; Hu et al., 2020; Li et al., 2019b; Berecibar et al., 2016). A significant research gap, often addressed with techniques like domain adaptation (Zhao et al., 2024), remains in understanding how these architectures perform on stochastic, real-world operational data.

A significant step in bridging this lab-to-real gap was recently made by (Liu et al., 2025) using the same 'IVST-EV' dataset central to our study. They proposed a multi-modal deep learning framework that fuses three types of inputs: 2D cell voltage maps, 1D feature sequences, and a set of 15 engineered point features. Their work demonstrates that a *hybrid* approach—combining domain-specific feature engineering with a deep ResNet architecture—can yield state-of-the-art performance. Our work, however, instead of designing a better hybrid model, we investigate the underlying assumption of the end-to-end paradigm itself, which forms the basis for representation learning (Bengio et al., 2013). We test whether deep models can succeed **without** the aid of such pre-engineered features,

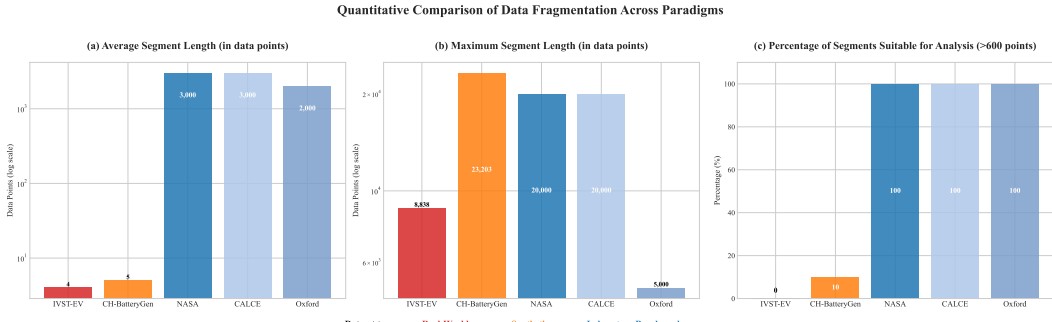

Figure 1: **The Stark Reality of Hyper-Fragmentation in Real-World Data.** This figure quantitatively demonstrates the fundamental incompatibility between our real-world 'IVST-EV' dataset and standard laboratory benchmarks. A *continuous operational segment* is defined as an uninterrupted period of vehicle operation (either charging or driving). We compare three key metrics: **(a) Average Segment Length:** On average, a real-world data segment from 'IVST-EV' contains only 4 data points, which is orders of magnitude smaller than the thousands of points found in typical laboratory cycles. This is a direct result of stochastic, real-world usage patterns. **(b) Maximum Segment Length:** Even the longest continuous segment in the 'IVST-EV' data is significantly shorter than in premier lab datasets like NASA's. While the Oxford dataset also has shorter cycles, its data consists entirely of complete, usable segments, unlike 'IVST-EV'. **(c) Effective Segments for Analysis:** This is the critical consequence. The percentage of segments long enough for conventional analysis (e.g., >600 points) is effectively zero in our data, compared to 100% in all laboratory settings. This necessitates the novel, fragmentation-robust methodologies developed in this work.

thereby probing the limits of autonomous learning with architectures like Neural Rough Differential Equations (Morrill et al., 2024) in this challenging data environment.

## 2.2 FEATURE ENGINEERING IN PROGNOSTICS AND HEALTH MANAGEMENT (PHM)

Parallel to the end-to-end learning paradigm, the field of Prognostics and Health Management (PHM) has a rich history rooted in signal processing and statistical feature engineering (Ng et al., 2020; Fink et al., 2020). The core philosophy is that domain expertise can guide the extraction of a small set of features with high "information density," effectively summarizing a system's health state from high-dimensional raw data.

In the context of batteries, engineered features often include Incremental Capacity Analysis (ICA) peaks or voltage curve plateaus (Dubarry et al., 2012; Birkl et al., 2017; Ye et al., 2022). The challenge is that many of these features also rely on stable conditions found in laboratory cycles. Other data-driven approaches have shown success in predicting battery lifetime from early-cycle data, again often in controlled settings (Severson et al., 2019; Paulson et al., 2022).

Our work contributes to this lineage by proposing a novel set of statistical features—the higher-order moments of lifetime voltage and current distributions—that are inherently robust to the "hyper-fragmentation" of our dataset. We demonstrate that this carefully designed, low-dimensional representation retains more relevant information for SOH prediction than the high-dimensional representations autonomously learned by deep models.

## 3 METHODOLOGY

Our methodology is designed as a direct, head-to-head comparison between the feature engineering paradigm and the end-to-end learning paradigm. Both approaches originate from the same preprocessed data and utilize the same meticulously engineered SOH labels.

### 3.1 DATA AND PREPROCESSING

Our primary dataset is `IVST-EV` (Liu et al., 2025), a large-scale collection of operational data from 300 vehicles, chosen for its realistic, non-laboratory conditions. A supplementary dataset,

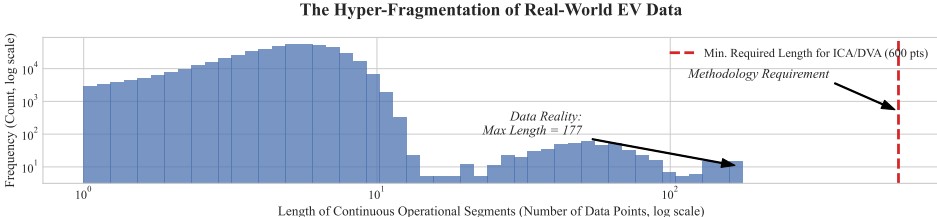

Figure 2: **The Hyper-Fragmentation of Real-World EV Data.** This log-log histogram visualizes the distribution of continuous operational segment lengths from the 'IVST-EV' dataset. The vast majority of segments are extremely short, with a maximum observed length of only 177 data points. The vertical dashed red line indicates the minimum length (e.g., 600 points) typically required for conventional prognostic methods like Incremental Capacity Analysis (ICA). This plot quantitatively demonstrates the core challenge of our dataset: a fundamental incompatibility between the reality of the fragmented operational data and the requirements of traditional analysis techniques, motivating the need for novel, fragmentation-robust methodologies.

`CH-BatteryGen` (China Automotive Engineering Research Institute and Huawei, 2025) (N=500), consisting of synthetic, well-structured cycles, was used to validate the robustness of our SOH labeling algorithms across different data types. The raw data, comprising high-frequency time-series measurements, underwent a rigorous preprocessing pipeline including cleaning, outlier clipping, and parsing of complex string-encoded sensor arrays into statistical summaries. The final clean data was stored in a columnar Parquet format for efficient access.

A key property of the 'IVST-EV' dataset, discovered during our initial data characterization, is its "hyper-fragmentation": the data consists of millions of short, disconnected operational segments, a direct consequence of real-world usage patterns. As quantitatively demonstrated in Figure 1, the statistical properties of these segments differ by orders of magnitude from standard laboratory benchmarks, posing a significant challenge for SOH label generation. This property, visually and quantitatively demonstrated in Figure 2, posed a significant challenge for SOH label generation.

### 3.2 SOH LABEL ENGINEERING

We investigated two primary SOH indicators: capacity-based ($SOH_C$) and internal resistance-based ($SOH_R$).

**Failure of Capacity-based Labeling**: Traditional Coulomb counting methods for $SOH_C$ were found to be inapplicable to the 'IVST-EV' dataset due to the lack of long, continuous charging segments. This resulted in only 11 out of 300 vehicles yielding a valid $SOH_C$ label, confirming this as an infeasible path for this dataset.

**Success of Resistance-based Labeling**: To overcome this, we developed a robust algorithm based on statistical regression over voltage-current steps to estimate an effective internal resistance. This method proved resilient to data fragmentation and successfully generated a consistent '$SOH_R$' indicator for all 300 'IVST-EV' vehicles.

**Normalization**: Finally, all derived SOH indicators were normalized to the $[0, 1]$ range using a "Symmetric Statistical Normalization" technique, where values were scaled based on the 5th and 95th percentiles of the entire dataset's distribution. This created the final target label for our primary experiments, '$SOH_{R\_Stat\_Norm}$'.

### 3.3 COMPETING PARADIGMS FOR SOH ESTIMATION

**Approach A: Feature Engineering + Traditional Machine Learning.** This paradigm tests the efficacy of a low-dimensional, domain-informed feature set.

Feature Vector: We engineered a novel 4-dimensional statistical "signature vector" for each vehicle, designed to be robust to fragmentation by capturing the global shape of the lifetime voltage and current distributions: $X_{FE} = [V_{\text{skew}}, V_{\text{kurtosis}}, I_{\text{skew}}, I_{\text{kurtosis}}]$. The skewness and kurtosis of the voltage and current distributions capture subtle changes in the battery's electrochemical behavior that manifest over its lifetime. For instance, as a battery degrades, its voltage response to load may become

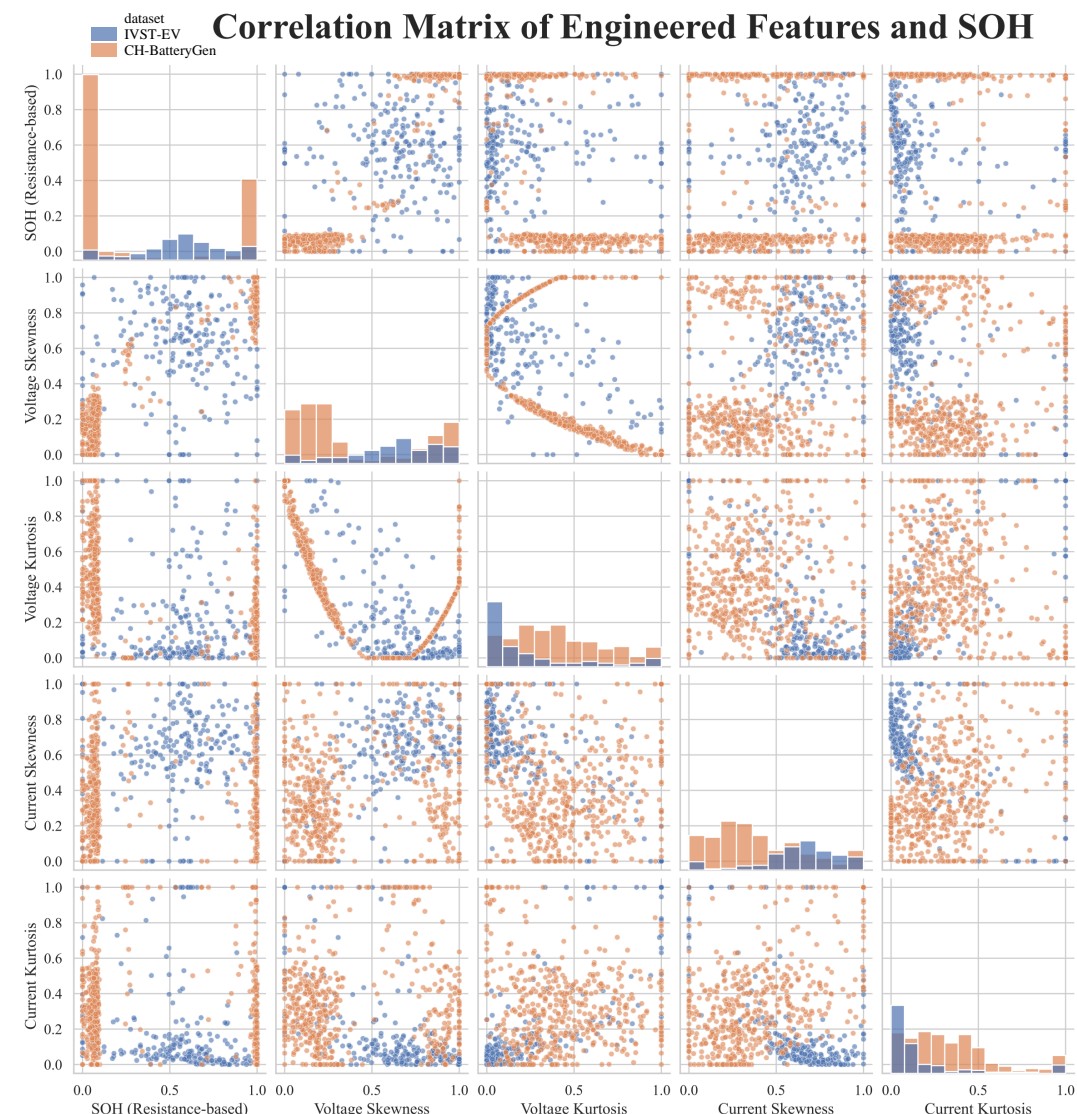

Figure 3: **Visualizing the Predictive Power of Engineered Statistical Features.** This figure presents a correlation matrix (pairplot) to explore the relationships between our four engineered statistical features and the target '$SOH_R$'. **Diagonal Panels:** Each panel on the diagonal displays the distribution (histogram) of a single variable, showing the range and frequency of its values across all 728 valid samples. **Off-Diagonal Panels:** Each off-diagonal panel is a scatter plot showing the relationship between two variables. The variable on the y-axis is determined by its row, and the variable on the x-axis is determined by its column. The data points are colored by their source dataset. **Key Insight:** The final row is the most critical for interpretation, as it explicitly plots each of the four statistical features (on the x-axes) against the '$SOH_R$' (on the y-axis). The clear, non-random trends visible in these plots (e.g., the relationship between 'Voltage Skewness' and SOH) provide strong visual evidence that our engineered features are highly correlated with battery degradation. This justifies their selection and explains the strong performance of the traditional machine learning models detailed in Section 4.

less symmetric (affecting skewness) and exhibit more extreme values (affecting kurtosis), providing a robust statistical fingerprint of its health state. The strong predictive potential of this feature set is visually demonstrated in Figure 3, which shows clear correlations between these features and the target SOH.

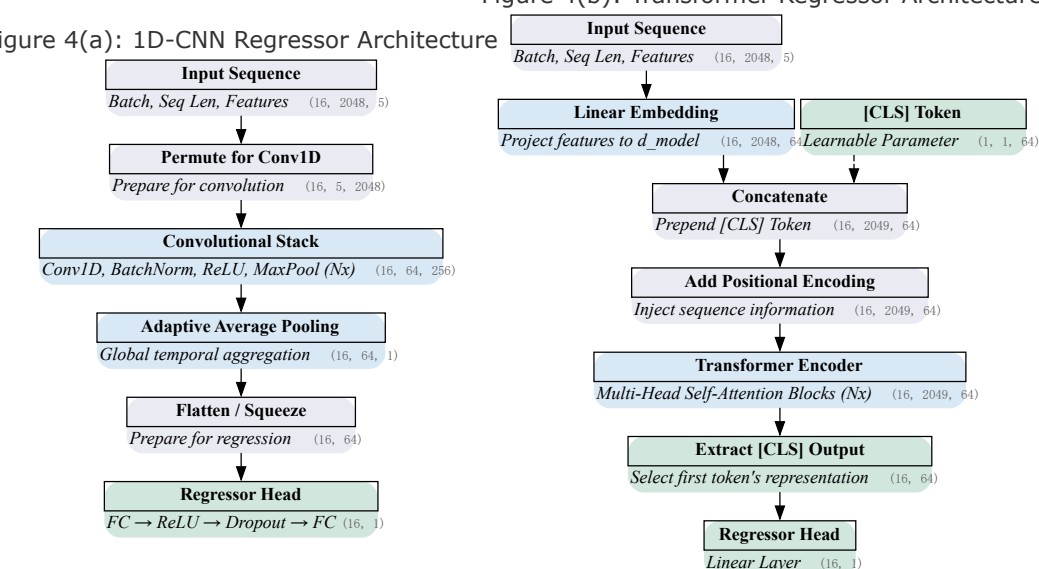

Figure 4: **Architectures of Key End-to-End Deep Learning Models.** This figure details the architectures of the two key models from our end-to-end deep learning approach. **(a)** The architecture of our best-performing 1D-CNN model. It processes the input sequence through a stack of convolutional blocks, followed by an adaptive pooling layer that ensures a fixed-size output for the final regressor head, making the architecture robust to variable input lengths. **(b)** The architecture of the Transformer model, which represents the state-of-the-art paradigm. Our implementation utilizes a learnable '[CLS]' token that is prepended to the input sequence. The final representation of this token, after being processed by the multi-head self-attention encoder blocks, is then used exclusively for the final SOH prediction.

Models Tested: A comprehensive suite of nine traditional models, including Logistic Regression, Support Vector Machines, Random Forest, XGBoost, LightGBM, and CatBoost (Chen & Guestrin, 2016; Ke et al., 2017; Prokhorenkova et al., 2018)..

**Approach B: End-to-End Deep Learning.**

This paradigm tests the ability of deep models to autonomously learn representations from raw data.

- **Input Data**: Raw, normalized time-series sequences of 5 features ('totalvoltage', 'totalcurrent', 'soc', 'maxtemperaturevalue', 'mintemperaturevalue') padded or truncated to a fixed length of 2048 timesteps.

- **Models Tested**: A diverse set of seven modern deep learning architectures whose implementations were pre-validated for structural correctness: 1D-CNN, LSTM, GRU, Bi-LSTM, a hybrid CNN-LSTM, LSTM with Attention, and a Transformer Encoder with a CLS token for regression. The detailed architectures for our best-performing model (1D-CNN) and the representative Transformer model are illustrated in Figure 4.

### 3.4 EVALUATION PROTOCOL

To ensure fair and robust comparison, we employed a multi-stage evaluation process. The traditional models in Approach A were evaluated via 5-fold repeated experiments to establish a statistically significant baseline. For the computationally intensive deep learning models in Approach B, we first conducted a broad exploratory scan of all 28 primary experimental configurations, followed by a 5-fold repeat experiment on the top-performing configurations to ensure statistical robustness. The primary evaluation metric is the Mean Squared Error (MSE), which is also converted to an estimated Coefficient of Determination ($R^2$) for comparison.

## 4 EXPERIMENTS AND RESULTS

Our experimental evaluation is structured to provide a clear and definitive answer to our central research question regarding the limits of the end-to-end paradigm on hyper-fragmented data. We first establish a baseline using the traditional feature engineering approach. We then present the complete results of our two-phase end-to-end deep learning exploration, culminating in a direct comparison between the champion of each paradigm. This approach allows us to quantify the performance gap that complements the findings from hybrid models such as the one proposed by Liu et al. (2025).

### 4.1 EXPERIMENTAL SETUP

The traditional machine learning baseline (Approach A) was evaluated on a combined dataset of 728 vehicles to ensure its statistical robustness. All end-to-end deep learning experiments (Approach B) were strictly conducted on the more challenging 'IVST-EV' dataset (N=300 for $SOH_R$, N=11 for $SOH_C$) to directly test their performance on real-world, fragmented data. The primary evaluation metrics are Mean Squared Error (MSE) and the estimated Coefficient of Determination ($R^2$).

### 4.2 BASELINE PERFORMANCE WITH FEATURE ENGINEERING

We first established the performance ceiling using our engineered 4-dimensional statistical feature vector. Each of the nine traditional models was trained and evaluated five times. The results, summarized in Table 1, demonstrate that the CatBoost model achieves a remarkably strong and stable performance, establishing a powerful benchmark for the subsequent deep learning experiments.

Table 1: Performance of traditional machine learning models trained on the 4-dimensional statistical feature vector to predict '$SOH_R$'. Results are the mean and standard deviation over 5 runs. The CatBoost model establishes the performance benchmark.

| Model | N Samples | Best Valid MSE ($\mu \pm \sigma$) | $R^2$ ($\mu \pm \sigma$) |
|---|---|---|---|
| **CatBoost** | **728** | $\approx 0.029$ | $0.8025 \pm 0.0227$ |
| RandomForest | 728 | $\approx 0.031$ | $0.7902 \pm 0.0251$ |
| LightGBM | 728 | $\approx 0.032$ | $0.7856 \pm 0.0234$ |
| SVR (RBF) | 728 | $\approx 0.034$ | $0.7722 \pm 0.0309$ |

### 4.3 END-TO-END DEEP LEARNING PERFORMANCE

Our evaluation of the end-to-end deep learning paradigm followed a two-phase protocol: a broad exploratory scan to assess all configurations, followed by a statistical validation of the most promising candidates.

#### 4.3.1 PHASE 1: BROAD EXPLORATORY SCAN

To begin, we conducted a single training run for each of the 28 primary experimental configurations. This scan served to map the entire performance landscape and identify candidates for more rigorous testing. The complete, unabridged results of this scan are presented in Table 2. Two key findings immediately emerged: the general infeasibility of $SOH_C$ prediction due to extreme data scarcity (N=11), and the superior potential of $SOH_R$ prediction (N=300).

#### 4.3.2 PHASE 2: STATISTICAL VALIDATION AND FINAL COMPARISON

Based on the exploratory scan, we selected the top-5 performing configurations for a more rigorous 5-fold repeat validation. This phase yielded two critical findings.

First, the anomalously good results for $SOH_C$ prediction were proven to be statistical artifacts. The 'LSTM+Attention' model, for instance, produced a mean MSE of $0.272035 \pm 0.088289$ over five runs, demonstrating that its single-run low MSE was an outlier caused by a "lucky" train-validation split on the insufficient dataset (N=11).

Second, we were able to establish a statistically robust performance benchmark for the end-to-end paradigm on the $SOH_R$ task. The 1D-CNN architecture consistently emerged as the most effective

Table 2: Complete, unabridged results of the single-run broad exploratory scan. This scan informed the selection of the top-5 candidates (highlighted in bold) for the subsequent statistical validation phase. Note the anomalously low MSE for two $SOH_C$ experiments, which were later investigated.

| Target (Y) | Input (X) | Model (f(X)) | N Samples | Best Valid MSE |
|---|---|---|---|---|
| $SOH_C$ | **Discharging Sequence** | **LSTM+Attention** | **11** | **0.009889** |
| $SOH_C$ | **Charging Sequence** | **Transformer** | **11** | **0.030570** |
| $SOH_R$ | **Charging Sequence** | **1D-CNN** | **300** | **0.050279** |
| $SOH_R$ | **Discharging Sequence** | **LSTM** | **300** | **0.054838** |
| $SOH_R$ | **Discharging Sequence** | **1D-CNN** | **300** | **0.057144** |
| $SOH_R$ | Discharging Sequence | GRU | 300 | 0.064018 |
| $SOH_C$ | Charging Sequence | LSTM | 11 | 0.066271 |
| $SOH_R$ | Charging Sequence | LSTM | 300 | 0.067705 |
| $SOH_R$ | Charging Sequence | GRU | 300 | 0.070909 |
| $SOH_R$ | Discharging Sequence | Transformer | 300 | 0.073043 |
| $SOH_R$ | Charging Sequence | Transformer | 300 | 0.073371 |
| $SOH_C$ | Full Cycle | Transformer | 11 | 0.081124 |
| $SOH_C$ | Discharging Sequence | GRU | 11 | 0.086339 |
| $SOH_C$ | Discharging Sequence | Transformer | 11 | 0.091862 |
| $SOH_C$ | Discharging Sequence | LSTM | 11 | 0.092075 |
| Multi | Charging Sequence | Transformer | 11 | 0.095409 |
| $SOH_C$ | Discharging Sequence | Bi-LSTM | 11 | 0.111080 |
| $SOH_C$ | Charging Sequence | 1D-CNN | 11 | 0.130776 |
| $SOH_C$ | Full Cycle | LSTM | 11 | 0.144465 |
| $SOH_C$ | Charging Sequence | LSTM+Attention | 11 | 0.155312 |
| $SOH_C$ | Charging Sequence | Bi-LSTM | 11 | 0.155859 |
| $SOH_C$ | Charging Sequence | CNN-LSTM | 11 | 0.178079 |
| Multi | Charging Sequence | LSTM | 11 | 0.191958 |
| $SOH_C$ | Discharging Sequence | 1D-CNN | 11 | 0.215876 |
| $SOH_C$ | Full Cycle | GRU | 11 | 0.239018 |
| $SOH_C$ | Discharging Sequence | CNN-LSTM | 11 | 0.247431 |
| Multi | Discharging Sequence | LSTM | 11 | 0.367730 |
| $SOH_C$ | Charging Sequence | GRU | 11 | 0.381148 |

model. Table 3 presents the final, statistically validated performance of the top-performing deep learning models and provides a direct comparison with the traditional feature engineering baseline. The performance gap is both significant and conclusive.

As visually summarized in Figure 5, the feature engineering paradigm not only achieves superior performance but does so with an input complexity that is orders of magnitude lower than the end-to-end approach.

Table 3: Final performance showdown. The table contrasts the statistically validated performance of the top-3 end-to-end deep learning models against the traditional CatBoost baseline. The traditional model, using only 4 engineered features, decisively outperforms the deep learning models operating on over 10,000 raw data points per sample ($2048 \times 5$).

| Methodology | Model | Input Representation | MSE ($\mu \pm \sigma$) | Final $R^2$ |
|---|---|---|---|---|
| **Feature Engineer** | **CatBoost** | **4 Statistical Features** | $\approx 0.029$ | $\approx 0.80$ |
| End-to-End DL | 1D-CNN | Raw Charging Sequence | $0.058679 \pm 0.009390$ | $\approx 0.12$ |
| | 1D-CNN | Raw Discharging Sequence | $0.059309 \pm 0.010618$ | $\approx 0.11$ |
| | LSTM | Raw Discharging Sequence | $0.060898 \pm 0.012399$ | $\approx 0.08$ |

## 5 CONCLUSION AND DISCUSSION

In this work, we conducted a rigorous comparison between feature engineering and end-to-end learning for SOH estimation on a noisy, hyper-fragmented, real-world EV dataset. Our results present

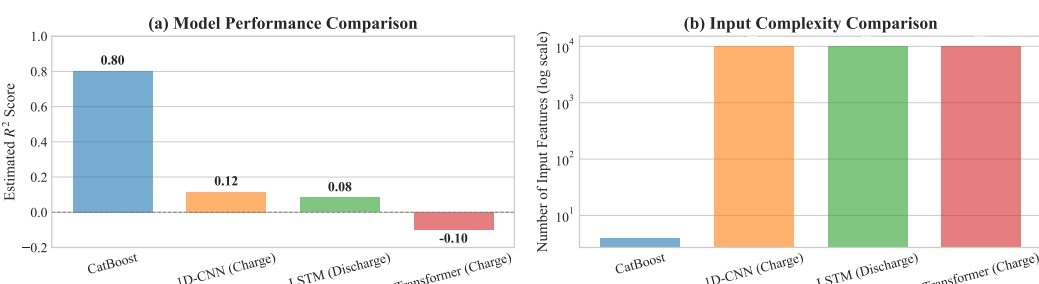

Figure 5: **The Decisive Superiority of Feature Engineering.** This two-panel figure provides the ultimate comparison between the two competing paradigms. **(a)** The performance comparison shows that the CatBoost model, trained on our 4-dimensional engineered feature vector, achieves a significantly higher $R^2$ score ($\approx 0.80$) than the best-performing end-to-end deep learning models. **(b)** The complexity comparison, plotted on a logarithmic scale, starkly illustrates the cost of this performance. The feature engineering approach requires only 4 input features, whereas the deep learning models operate on over 10,000 raw data points ($2048 \times 5$). Taken together, the figure demonstrates that our feature engineering approach achieves dramatically superior performance at a fraction of the input complexity.

a clear and compelling conclusion: a traditional CatBoost model trained on a mere four, domain-knowledge-driven statistical features ($R^2 \approx 0.80$) outperformed a wide array of sophisticated deep learning architectures, whose best statistically validated performance was only $R^2 \approx 0.12$.

This finding offers a critical counterpoint and a deeper insight into the state-of-the-art results established on this dataset. For instance, Liu et al. (2025) achieved high accuracy using a complex, multi-modal deep learning model. However, their model's success was heavily reliant on an extensive set of pre-engineered features, including 2D feature maps and 15 distinct point-based health indicators. Our work reveals that the "secret sauce" in this data regime may not be the complexity of the deep learning architecture itself, but rather the quality and information density of the features provided to it. By isolating and stress-testing the pure end-to-end paradigm, we demonstrate its fundamental limitations when faced with hyper-fragmented signals.

This "Triumph of Feature Engineering," aligns with a growing body of work that highlights the continued importance of interpretable, domain-informed models in real-world applications (Wen et al., 2024; Hu et al., 2025). The significant performance gap suggests that in our high-noise, data-constrained setting, the "information density" captured by our simple statistical features was far more potent than what current deep learning models could autonomously extract from raw data. This underscores the value of interpretable representations, a critical aspect for safety-critical systems as highlighted in recent reviews on explainable AI (Wang & Chen, 2024).

Our work does not diminish the potential of deep learning, but rather refines our understanding of its application boundaries. It suggests that for certain industrial AI challenges, the optimal path may not be to rely solely on scaling larger end-to-end models, but to invest in creating robust, information-rich features. Future work could explore hybrid approaches, potentially integrating our robust statistical features with physics-informed neural networks (Gao et al., 2024) or advanced domain adaptation techniques (Zhao et al., 2024) to further bridge the lab-to-real gap. Moreover, emerging architectures such as Neural Rough Differential Equations (Morrill et al., 2024) or even Large Language Model frameworks (Yunusoglu et al., 2025) may offer new pathways for modeling such complex industrial time-series. Ultimately, our study contributes to the foundational discussion on representation learning, emphasizing that the optimal representation is highly context-dependent (Bengio et al., 2013).

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

## ETHICS STATEMENT

The research presented in this paper adheres to the ICLR Code of Ethics. Our study is based on two core datasets, for which we have carefully considered the ethical implications.

Our primary dataset, 'IVST-EV', is a large-scale collection of real-world operational data from electric vehicles. This dataset was formally introduced and made publicly available for academic research purposes by Liu et al. (2025). We acknowledge the ethical responsibilities associated with using such data. According to the original data providers, the dataset was fully anonymized prior to its release, with all potential personally identifiable information (PII)—including but not limited to Vehicle Identification Numbers (VINs) and precise GPS location data—rigorously removed to protect the privacy of all vehicle owners. Our use of this established public research dataset is in full accordance with the terms of its release.

Our supplementary dataset, 'CH-BatteryGen' (China Automotive Engineering Research Institute and Huawei, 2025), was used for validation purposes. As this dataset is synthetically generated by an AI model and contains no real-world personal or operational data, it does not raise privacy or ethical concerns.

The objective of our work is to enhance the safety and reliability of battery management systems, contributing positively to a more sustainable transportation ecosystem. We foresee no direct negative societal consequences resulting from this research.

## REPRODUCIBILITY STATEMENT

We are committed to ensuring the reproducibility of our research. To facilitate this, we provide comprehensive details regarding our algorithms, datasets, experimental configurations, and computational environment in the main paper and a detailed appendix. All figures presented in this paper were generated using scripts that process our final experimental results, ensuring a direct and verifiable link between our findings and their visualization.

**Algorithms and Models** The conceptual foundation of our novel methodologies is provided as high-level pseudocode in Appendix A.1. This includes the detailed algorithms for our robust resistance-based SOH label engineering (Algorithm 1), the 4D statistical feature extraction (Algorithm 2), and the symmetric statistical normalization (Algorithm 3). Furthermore, a comprehensive table detailing the architectures and hyperparameters for all traditional and deep learning models is available in Appendix A.2, ensuring that our model configurations can be precisely replicated.

**Datasets and Preprocessing** Our study relies on two datasets. The primary 'IVST-EV' dataset (Liu et al., 2025) is proprietary due to commercial agreements and cannot be publicly released. However, to ensure maximum transparency, we provide a detailed statistical characterization of its "hyper-fragmentation" property in Section 3 (Figures 1 and 2). The complete data preprocessing pipeline, from raw CSV parsing to the final Parquet format, is detailed in Appendix A.3. The supplementary 'CH-BatteryGen' dataset (China Automotive Engineering Research Institute and Huawei, 2025), used for validation, is publicly available.

**Computational Environment and Rigor**   All experiments were conducted in a Python 3.9 environment. The key libraries used include PyTorch (v1.12), scikit-learn (v1.1), CatBoost (v1.0), and Pandas (v1.4). The deep learning models were trained on a single NVIDIA RTX3060ti 16GB GPU. To ensure the statistical significance of our main findings, both the traditional machine learning baseline (Table 1) and the top-performing deep learning models (Table 3) were evaluated over 5 repeated runs using different random seeds for data splitting.

# A   APPENDIX

## A.1   CORE ALGORITHMS IN PSEUDOCODE

This section provides the pseudocode for the three core custom algorithms developed in this study. Each algorithm is preceded by a comprehensive explanation of its purpose, methodology, and significance to the paper's overall contribution. This structure is designed to ensure both conceptual clarity and technical reproducibility.

**Algorithm 1: Robust Resistance-based SOH Label Engineering**   The primary challenge in using the 'IVST-EV' dataset is the absence of complete charge-discharge cycles, which renders traditional Coulomb counting for capacity-based SOH labels ($SOH_C$) infeasible (as shown in Section 3.2, with a success rate of only 11/300). To overcome this, we developed the robust algorithm detailed below. Its methodology is to first isolate data points within a statistically stable State-of-Charge (SOC) plateau (40-60%), where the relationship between voltage and current is most representative of the battery's internal state. It then calculates the instantaneous ohmic resistance for thousands of such points across the vehicle's lifetime and takes the median value. The significance of this algorithm is foundational: it provides a reliable, noise-resistant, and consistently computable internal resistance-based SOH label ($SOH_R$) for every vehicle in the dataset, thereby creating the high-quality ground truth upon which our entire study is built.

---

**Algorithm 1** Robust Resistance-based SOH Label Engineering

---

1: **Input:** A vehicle's entire lifetime data `D_vehicle`.
2: **Output:** A single scalar value `SOH_R_Stat` (in mΩ).
3: **function** GETRESISTANCELABEL(`D_vehicle`)
4:      ▷ Filter for the stable SOC plateau.
5:     `D_stable` ← `D_vehicle`.filter( $40 \leq$ `D_vehicle`['soc'] $\leq 60$ )
6:      ▷ Filter for points with meaningful current.
7:     `D_valid_ir` ← `D_stable`.filter( |`D_stable`['current']| $> 1.0$ A )
8:     **if** count(`D_valid_ir`) ¡ 10 **then**
9:         **return** None
10:     **end if**
11:      ▷ Calculate instantaneous resistance for all valid points.
12:     `R_points` ← (`D_valid_ir`['voltage']/|`D_valid_ir`['current']|) $\times 1000$
13:      ▷ Use the median for a robust estimate against outliers.
14:     `SOH_R_Stat` ← Median(`R_points`)
15:     **return** `SOH_R_Stat`
16: **end function**

---

**Algorithm 2: 4D Statistical Feature Vector Extraction**   This algorithm details the construction of our novel 4-dimensional feature vector, which is the cornerstone of our successful feature engineering approach. The core purpose is to create a compact, yet information-rich, "fingerprint" of a vehicle's entire operational history that is inherently robust to the "hyper-fragmentation" of the data. The methodology involves calculating the third and fourth order statistical moments—skewness and kurtosis—for the global distributions of the lifetime voltage and current signals. The significance of this approach is its remarkable effectiveness: these four simple statistical values, capturing the overall shape and asymmetry of the battery's electrical behavior, proved to contain more predictive power for SOH than the high-dimensional representations learned by complex end-to-end deep learning models.

---

**Algorithm 2** 4D Statistical Feature Vector Extraction

---

1: **Input:** A vehicle's entire lifetime data `D_vehicle`.
2: **Output:** A 4-dimensional feature vector $X_{FE}$.
3: **function** EXTRACTSTATISTICALFEATURES(`D_vehicle`)
4:                                                                            ▷ Extract the complete time-series for voltage and current.
5:     $\mathbf{V} \leftarrow$ `D_vehicle`['voltage']
6:     $\mathbf{I} \leftarrow$ `D_vehicle`['current']
7:                                                               ▷ Calculate 3rd (skewness) and 4th (kurtosis) order moments.
8:     `v_skew` $\leftarrow$ Skewness($\mathbf{V}$)
9:     `v_kurtosis` $\leftarrow$ Kurtosis($\mathbf{V}$)
10:    `i_skew` $\leftarrow$ Skewness($\mathbf{I}$)
11:    `i_kurtosis` $\leftarrow$ Kurtosis($\mathbf{I}$)
12:                                                                                       ▷ Assemble the final feature vector.
13:    $X_{FE} \leftarrow$ [`v_skew`, `v_kurtosis`, `i_skew`, `i_kurtosis`]
14:    **return** $X_{FE}$
15: **end function**

---

**Algorithm 3: Symmetric Statistical Normalization**   A consistent normalization scheme is critical for the fair comparison of models and for stable model training. This algorithm implements the "Symmetric Statistical Normalization" technique used throughout our study. Its methodology avoids using fixed physical anchors (which may not be known) and instead establishes a robust data-driven scale. It defines the "healthiest" state (SOH=1.0) and "unhealthiest" state (SOH=0.0) using the 5th and 95th percentiles of the entire fleet's data distribution for a given metric. The significance of this method is twofold: first, it ensures that all features and labels are scaled to a consistent [0, 1] range in a way that is robust to extreme outliers. Second, it correctly handles the physical meaning of different metrics by applying a reversed scale for indicators like resistance, where a higher raw value corresponds to a lower state of health.

---

**Algorithm 3** Symmetric Statistical Normalization

---

1: **Input:** A value 'val' to normalize, the list of all values '$all_vals$' from the fleet, a boolean '$reverse_scale$'.
2: **Output:** A normalized value `val_norm` in the range $[0, 1]$.
3: **function** NORMALIZEVALUE('val', `all_vals`, `reverse_scale`)
4:                                                    ▷ Establish anchors using 5th and 95th percentiles of the population.
5:     `min_anchor` $\leftarrow$ Quantile(`all_vals`, 0.05)
6:     `max_anchor` $\leftarrow$ Quantile(`all_vals`, 0.95)
7:     **if** `max_anchor` - `min_anchor` $\approx 0$ **then**
8:         **return** 0.5                                                                            ▷ Handle cases with no variance.
9:     **end if**
10:                                                                      ▷ Apply normalization with optional scale reversal.
11:    **if** `reverse_scale` is True **then**
12:        $soh \leftarrow ($`max_anchor` $-$ val$)/($`max_anchor` $-$ `min_anchor`$)$
13:    **else**
14:        $soh \leftarrow ($val $-$ `min_anchor`$)/($`max_anchor` $-$ `min_anchor`$)$
15:    **end if**
16:                                                                      ▷ Clip the final value to the standard [0, 1] range.
17:    `val_norm` $\leftarrow$ Clip($soh$, 0.0, 1.0)
18:    **return** `val_norm`
19: **end function**

---

## A.2   MODEL HYPERPARAMETERS AND ARCHITECTURE DETAILS

This section provides a comprehensive overview of the hyperparameters and architectural configurations for all models evaluated in this study. We separate the models into two categories: the traditional machine learning models used in our feature engineering approach, and the end-to-end

deep learning models. All configurations were kept consistent across relevant experiments to ensure a fair and robust comparison.

**Traditional Machine Learning Models**   The traditional models, evaluated in Section 4.2, were implemented using popular libraries such as scikit-learn, XGBoost, LightGBM, and CatBoost. For most models, we utilized the default hyperparameters provided by the respective libraries, as our primary goal was to establish a robust baseline rather than perform exhaustive hyperparameter tuning. Key non-default parameters and settings are listed in Table 4. All models were trained on features scaled by a 'StandardScaler'.

Table 4: Hyperparameters for Key Traditional Machine Learning Models.

| Model | Key Hyperparameters / Settings |
|---|---|
| **CatBoost** | 'iterations=1000', 'learning rate=0.03', 'depth=6', 'verbose=0', 'random state=42' |
| **RandomForest** | 'n estimators=100', 'max depth=None', 'min samples split=2', 'random state=42' |
| **LightGBM** | 'n estimators=100', 'learning rate=0.1', 'num leaves=31', 'random state=42' |
| **XGBoost** | 'n estimators=100', 'learning rate=0.1', 'max depth=3', 'random state=42' |
| **SVR (RBF)** | 'kernel='rbf'', 'C=1.0', 'gamma='scale'' |

**End-to-End Deep Learning Models**   All deep learning models were implemented in PyTorch and trained under a unified experimental setup to ensure comparability. The detailed training configuration and model-specific architectures are presented in Table 5. These parameters correspond to the models whose architectures are illustrated in Figure 4 and whose results are presented in Section 4.3.

Table 5: Training and Architectural Hyperparameters for End-to-End Deep Learning Models.

| Category | Parameter | Value |
|---|---|---|
| **General Training** | Optimizer | Adam |
| | Learning Rate | $1 \times 10^{-4}$ |
| | Batch Size | 16 |
| | Number of Epochs | 50 |
| | Loss Function | Mean Squared Error (MSE) |
| **Input Data Shape** | Sequence Length | 2048 timesteps |
| | Number of Features | 5 ('totalvoltage', 'totalcurrent', 'soc', 'maxtemp', 'mintemp') |
| | Normalization | Per-sample Z-score normalization |
| | Device | NVIDIA RTX3060ti 16GB GPU (CUDA) |
| **Recurrent Models (LSTM, GRU, Bi-LSTM)** | Hidden Dimension | 64 |
| | Number of Layers | 2 |
| | Dropout | 0.1 |
| | Bidirectional (for Bi-LSTM) | True |
| **1D-CNN Model** | Convolutional Blocks | 3 |
| | Kernel Sizes | [7, 5, 3] |
| | Output Channels | [16, 32, 64] |
| | Final Pooling Layer | 'AdaptiveAvgPool1d(1)' |
| **Transformer Model** | Hidden Dimension ($d_{model}$) | 64 |
| | Number of Encoder Layers | 2 |
| | Number of Attention Heads | 4 |
| | Dropout | 0.1 |
| | Regression Strategy | Learnable '[CLS]' token output |

## A.3   DATA PREPROCESSING PIPELINE

The raw data for each vehicle in the 'IVST-EV' dataset was provided as a single large CSV file. To prepare this data for our study, we executed a rigorous and consistent preprocessing pipeline for each vehicle, based on the logic implemented in our script '$step1_data_preprocess.py$'. This pipeline was designed to clean the data, handle anomalies, and convert the raw text-based format into a numerical format suitable for analysis. The key steps were as follows:

1. **Numerical Clipping:** To handle sensor noise and extreme outliers, key numerical columns were clipped to within their plausible physical ranges. The clipping bounds were: 'soc' (0, 100), 'speed' (0, 250 km/h), 'totalvoltage' (250V, 410V), and 'totalcurrent' (-400A, 200A). Any values outside these ranges were set to the respective boundary value.

2. **Linear Interpolation:** After clipping, any remaining missing values ('NaN') in the numerical columns were filled using linear interpolation. This step ensures data continuity, which is particularly important for time-series analysis, while avoiding the introduction of artificial biases.

3. **String Array Parsing:** A critical challenge of the raw dataset was the presence of high-frequency, cell-level data encoded as large, tilde-separated strings (e.g., "3.71 3.72 ..."). For each timestep, these strings (specifically for 'batteryvoltage' and 'probetemperatures') were parsed into numerical arrays. We then computed four key statistical summaries—mean, standard deviation, minimum, and maximum—for each array. These summaries were stored as new, separate columns (e.g., '$batteryvoltage_mean$', '$batteryvoltage_std$', etc.), effectively converting the unstructured text information into a structured numerical format.

4. **Finalization and Formatting:** After the statistical summaries were generated, the original large string columns were dropped to reduce the dataset's size and complexity. The final, cleaned, and fully numerical dataset for each vehicle was then saved in the efficient Parquet format, which served as the standardized input for all subsequent analysis steps described in this paper.

## A.4 Use of Large Language Models in Manuscript Preparation

During the preparation of this manuscript and the accompanying code, we utilized a large language model (LLM) as a writing and technical assistant to enhance the quality, clarity, and correctness of our submission. The specific applications of the LLM were as follows:

**Language Refinement:** The LLM was employed to polish the manuscript's language by improving sentence structure, ensuring grammatical correctness, and enhancing the overall readability and flow of the prose.

**Technical Formatting and Debugging:** The model served as a technical assistant for formatting complex tables in LaTeX. Additionally, it was used to help diagnose and suggest solutions for compilation errors encountered in both our experimental Python scripts and the LaTeX source code.

We affirm that the core intellectual contributions of this work are entirely our own. This includes the formulation of the research question, the design and execution of the experiments, the interpretation and analysis of the results, and the formulation of the final conclusions. The role of the LLM was strictly limited to that of a productivity and polishing tool.

