# OpenReview forum: "On the Limits of End-to-End Learning: Why Statistical Features Dominate in Hyper-Fragmented Battery Prognostics"
_ICLR.cc/2026/Conference — ICLR 2026 Conference Withdrawn Submission_

### Official Review · Reviewer_Jevm · 2025-10-30

**Soundness:** 1
**Presentation:** 3
**Contribution:** 1
**Rating:** 0
**Confidence:** 4

**Summary:**

This work presents an empirical evaluation of two learning paradigms to predict the State-of-Health (SOH) of batteries in electric vehicles. (A) Traditional machine learning and (B) end-to-end deep learning. The traditional approaches take as input four statistical features (skewness and kurtosis of both voltage and current), after which traditional machine learning is applied (e.g., CatBoost). The deep learning approaches take as input the raw data and learn a representation internally. The results indicate that the traditional approaches based on statistical features outperform the deep-learning approach.

**Strengths:**

S1 [Originality] An interesting problem question is posed: the effect of feature engineering for predicting SOH.

S3 [Clarity] The authors provide many details regarding the processing of the data.

S4 [Significance] The paper shows that one should not blindly trust that deep learning methods will prevail, and it might be useful to invest some time in analysing more traditional algorithms.

**Weaknesses:**

The weaknesses will come back in my questions below. Below I summarise the main weaknesses and refer to the related questions.

W1. There are many inconsistencies within the paper. For example, the maximum length of a sequence in IVST-EV (see Q2), the number of used traditional models (Q5)

W2. There are multiple questionable choices within the experimental setup with insufficient argumentation (Q3, Q4, Q6, Q8, Q9).

W3. Overall, the contributions of this work are very limited. The authors scope down their work significantly, e.g., no hyperparameter tuning (Q12) or limited statistical features (Q13). However, these are important research questions which do not require significant amount effort nor compute to answer, but which may bring significant new insights.

**Questions:**

Q1 [line 106] The authors refer to “we investigate the underlying assumption of the end-to-end paradigm itself, which forms the basis for representation learning”. Can you elaborate? What exactly is this definition and how do you investigate it within this work?

Q2 [Figures 1 and 2] What is the maximum length of the sequences in IVST-EV? Because in Figure 1 (b) it is 8,838 while it is 177 in Figure 2.

Q3 [line 179-184] Did you experiment with other preprocessing steps? What was the effect of these? Because it could be that a specific form of preprocessing adds a positive or negative bias to any learning method.

Q4 [Section 3.2] Related to Q3. If you create a custom procedure for generating ground truth labels, don’t you also include some bias towards learning methods that can approximate this custom procedure? How did you prevent including such bias in your experiments?

Q5 [line 300] You claim to have used 9 traditional machine learning models but then only list 6 names. In the experimental results, you again mention 9 (line 342) while showing the results of only 5 algorithms (table 1). Similarly, table 4 in Appendix 2 only shows hyperparameters of 5 machine learning models.

Q6 [line 306] You truncate or pad the sequences to a length of 2048. Why did you pick this value? Because this seems an extremely large value for an average sequence length of 4 (figure 1a). In the average case, only 0.19% of the input sequence (4/2048) is relevant. This might also give an indication of the poor performance of the deep learning models.

Q7 [line 321] You mention that there are 28 configurations for deep learning methods, but you only use 7 models. Can you elaborate on this? I can count in table 2 that there are 28 rows, in which the input X, the target Y, and the model f(X) changes, but I don’t see all possible combinations. How did you select 28 configurations?

Q8 [line 336] Why did you evaluate the deep learning methods on a different set than the traditional models? Even more, this other set is “more challenging”, which would naturally lead to lower performance. Even more, if some deep learning models are trained using only N=11 instances, don’t you risk overfitting given that these models are known to be data-hungry?

Q9 [line 396] What do you mean by the “multi” target?

Q10 [line 480] Final line of the text: “The study contributes to the foundational discussion on representation learning, emphasizing that the optimal representation is highly context-dependent.” Can you elaborate on this? How did the presented work emphasize that the optimal representation is highly context-dependent?

Q11 [reproducibility statements] The number of details within the paper is good and suffice to reproduce the results as indicated in the statements. However, wouldn’t it make more sense to simply release the scripts, such that the community can directly use your scripts to reproduce the results without the risk of including any bugs? In this case you could also release the raw experimental results and formatted data.

Q12 [appendix 2] In this work, you did not include any hyperparameter tuning. What do you think would be the effect of tuning on the final conclusions? Do you expect that the performance of end-to-end learning improves with proper hyperparameter tuning?

Q13 [general] There are 2 standard statistics computed from 2 features for a total of 4 inputs. Did you also experiment with other features? If yes, which features and what was the result? If no, why not?
A more thorough analysis of the concrete features that were used is required to accurately answer the research question. Even more, it is not difficult to generate a large set of features, for example using tsfresh. You can either generate a simple set of basic features or a more comprehensive set and apply several feature selection strategies. Alternatively, it would also be possible to compute the Catch22 features and feat these to the traditional models. Such an analysis would more accurately indicate the performance of feature engineering, including the sensitivity of traditional machine learning to the used features.

Minor remarks:
- Line 179: You use the variable N without defining it.
- Line 258: I believe this should be first row instead of final row.
- Line 350: The third column of table 1 indicates \mu \pm \sigma, which is not shown.
- Line 804: Use \_ in latex to properly use underscores (also lines 819 and 820).

Incorrect use of opening quotes:
- Line 039: “lab-to-real”
-Line 140: “information density”
- Line 460: “secret sauce”
- Line 465: “triumph of feature engineering”
- Line 467: “information density”

---

### Official Review · Reviewer_fYf1 · 2025-10-30

**Soundness:** 2
**Presentation:** 1
**Contribution:** 1
**Rating:** 2
**Confidence:** 4

**Summary:**

This paper studies whether end-to-end deep learning can outperform compact, engineered statistical features for EV battery SOH on the IVST-EV dataset. The authors find that a 4-D feature vector (skew/kurtosis of voltage/current) with CatBoost substantially outperforms deep learning baselines in predictive accuracy (MSE/R²).

**Strengths:**

Research question is clear

**Weaknesses:**

- While the findings might be interesting for the battery community, this is too niche for ICLR. I advise the authors to add more intuition and put more emphasis on general takeaways beyond the IVST-EV dataset and bettery tasks if their aim at targeting a machine learning conference like ICLR; otherwise, I think the work would be a relevant contribution to specialized journals.
- Only one dataset, which was recently used, is used for the analysis. This raises the question of whether the results are generable to other datasets.
- The manuscript is unclear for people not familiar with IVST-EV, as many steps are dataset-specific.
- No detailed analysis is performed on whether the networks fail to converge or are overfitting the data.
- No comparison with strong, widely used time-series baselines and the hybrid approach of Liu et al. (2025).

**Questions:**

Feel free to address the points I presented in the weakness

---

### Official Review · Reviewer_pb16 · 2025-10-31

**Soundness:** 2
**Presentation:** 2
**Contribution:** 2
**Rating:** 2
**Confidence:** 4

**Summary:**

This paper compares feature engineering and end-to-end deep learning on battery prognostics. At first, the paper claimed the hyper-fragmentation issue in real-world data. Then, through 4 hand-crafted features, the paper constructed a strong baseline method for hand-crafted features battery prognostics. Extensive experiments and analysis demonstrate that the traditional machine learning method largely outperformed a series of deep learning methods.

**Strengths:**

- Comparing  feature engieering based traditional machine learning with deep learning is a very interesting topic.
- A good baseline is constructed for battery prognostics.

**Weaknesses:**

- The title, *Why Statistical Features Dominate in Hyper-Fragmented Battery Prognostics*, seems to dive into the reasons why statistical features dominates. However, the paper does not provide enough analysis or insights at that point.
-  As we all know, data amount means a lot when comparing deep learning methods with traditional machine learning. There seems only 300 samples, which certainly provide better conditions for traditional machine learning.
- Though the topic is good, the novelty of this paper is much lower than the acceptance line in my eyes. There is no novel methods or insightful analysis.

**Questions:**

See Weaknesses.

---

### Official Review · Reviewer_PpJG · 2025-11-01

**Soundness:** 3
**Presentation:** 3
**Contribution:** 3
**Rating:** 4
**Confidence:** 3

**Summary:**

This paper investigates the limits of end-to-end deep learning for electric vehicle (EV) battery State-of-Health (SOH) estimation in highly fragmented real-world operational data. Using the large-scale IVST-EV dataset, the authors compare two paradigms: (1) a traditional feature engineering pipeline with a CatBoost regressor trained on a compact 4D statistical feature vector, and (2) seven modern end-to-end deep learning architectures (CNN, LSTM, GRU, Transformer, etc.) trained directly on raw time-series inputs. The results show a striking gap in performance: the engineered-feature approach achieves an R² ≈ 0.80, while the best deep learning model only reaches ≈ 0.12. The study concludes that, under extreme data fragmentation, simple statistical features convey more predictive information than end-to-end learned representations.

**Strengths:**

- The paper tackles a rarely examined but crucial question — the boundary conditions under which end-to-end learning fails in real-world industrial settings. This “anti-hype” direction is novel and timely.

- The empirical design is rigorous, with consistent preprocessing, transparent pseudocode, and reproducibility details. The head-to-head comparison across paradigms is methodologically sound.

- The paper is well-organized and clearly written. Figures (especially Fig. 5) effectively illustrate both performance and input complexity.

- The findings challenge dominant assumptions in representation learning, providing valuable insights for industrial AI and PHM (Prognostics and Health Management). The study establishes a strong baseline and highlights an underexplored failure mode of deep models.

**Weaknesses:**

- Although the paper emphasizes transparency, the primary IVST-EV dataset is proprietary and not publicly available, limiting true reproducibility.

- Deep models are trained on truncated fixed-length sequences, potentially undermining their ability to exploit temporal dependencies. The lack of data augmentation, self-supervised pretraining, or domain adaptation may bias the comparison against deep learning.

- The paper could benefit from a qualitative exploration of why deep models underperform—e.g., via feature visualization or attention analysis—to support its conclusion more convincingly.

**Questions:**

- Have the authors considered self-supervised or physics-informed pretraining approaches (e.g., contrastive learning, PINNs) that might alleviate data fragmentation effects?

- What specific factors make the 4D feature vector so effective—does it generalize across different vehicle types or datasets?

- Would integrating these statistical features into deep architectures (as hybrid inputs) improve results? A short ablation or hybrid baseline could strengthen the paper’s claims.

- Since the proprietary dataset cannot be released, can the authors provide a small public subset or a synthetic equivalent to support reproducibility?

---

### Official Review · Reviewer_hTZH · 2025-11-01

**Soundness:** 3
**Presentation:** 3
**Contribution:** 2
**Rating:** 6
**Confidence:** 3

**Summary:**

The paper tests whether end-to-end deep models can learn SOH directly from short, irregular EV telemetry without domain features. It contrasts CatBoost on a compact 4-D statistical vector with several 1D-CNN/LSTM/Transformer baselines. On IVST-EV, the 4-D CatBoost attains about R², while the best end-to-end model is ~0.12; a transformer is below zero. The results suggest information-dense summaries, not architecture complexity, drive accuracy in this data regime.

**Strengths:**

1. Sharp, practically important question for real-world SOH from field data.
2. Uses a large, publicly released operational dataset rather than lab coin cells.
3. Clean experimental contrast; negative E2E results are clearly reported.
4. Figures 3 & 5 convincingly illustrate (i) engineered features carry signal and (ii) E2E suffers despite far greater input complexity.

**Weaknesses:**

1. it is unclear whether the 4 stats are computed strictly from information available at prediction time.
2. E2E is asked to predict a lifetime target from tiny fragments without explicit lifetime aggregation
3. the paper lacks time-aware baselines (T-LSTM, continuous-time encoders, delta-time embeddings)
4.  only R²; the experimental part needs calibration and thresholded accuracy around warranty-relevant cutoffs (e.g., within ±5% SOH, <80% SOH).

**Questions:**

1. Exactly how are the four statistics computed and aggregated at inference time? Do they ever use information from after the prediction timestamp?
2. Any results with time-aware encoders (delta-time embeddings, T-LSTM, Neural-ODE/continuous-time Transformers)?
3. Could you provide parameter counts, optimization settings, et al. ?
4. What is the effect of self-supervised pretraining (masked-reconstruction / contrastive across trips) on the E2E models?
5. How does CatBoost/XGBoost perform with a large auto-generated TS-feature library compared to the handcrafted 4-D vector?
6. Can you report calibration and thresholded metrics (±5% SOH; <80% SOH) to reflect operational risk?

---

### Note · Authors · 2025-11-12

**Comment:**

Dear Program Chair, Area Chairs, and Reviewers,

We are deeply grateful for your thorough review and invaluable feedback on our submission. We sincerely appreciate the time, expertise, and constructive criticism you dedicated to evaluating our work. Your insightful comments have significantly clarified areas requiring refinement, and we fully concur with your suggestions for strengthening the technical rigor and clarity of the paper.

In light of your recommendations, we are committed to revising the manuscript comprehensively to address all points raised. The revised version, incorporating every suggested improvement, will be submitted within the standard revision period.

Thank you for your exceptional guidance, which has been instrumental in elevating the quality and impact of our research. We are truly honored to have your expertise inform this important work.

With utmost respect

**Withdrawal Confirmation:**

I have read and agree with the venue's withdrawal policy on behalf of myself and my co-authors.